# Reclaiming Indigenous Planning as a Pathway to Local Water Security

**Robert J. Patrick [1],\*, Kellie Grant [2] and Lalita Bharadwaj [3]**

[1] Department of Geography and Planning, University of Saskatchewan, Saskatoon, SK S7N 5C5, Canada
[2] Planning and Development Division, Saskatoon, SK S7K 0J5, Canada; kellie.grant@saskatoon.ca
[3] School of Public Health, University of Saskatchewan, Saskatoon, SK S7N 2Z4, Canada;
lalita.bharadwaj@usask.ca
\* Correspondence: robert.patrick@usask.ca; Tel.: +1-306-966-6653

**Abstract:** Access to drinkable water is essential to human life. The consequence of unsafe drinking water can be damaging to communities and catastrophic to human health. Today, one in five First Nation communities in Canada is on a boil water advisory, with some advisories lasting over 10 years. Factors contributing to this problem stretch back to colonial structures and institutional arrangement that reproduce woefully inadequate community drinking water systems. Notwithstanding these challenges, First Nation communities remain diligent, adaptive, and innovative in their efforts to provide drinkable water to their community members. One example is through the practice of source water protection planning. Source water is untreated water from groundwater or surface water that supplies drinking water for human consumption. Source water protection is operationalized through land and water planning activities aimed at reducing the risk of contamination from entering a public drinking water supply. Here, we introduce a source water protection planning process at Muskowekwan First Nation, Treaty 4, Saskatchewan. The planning process followed a community-based participatory approach guided by trust, respect, and reciprocity between community members and university researchers. Community members identified threats to the drinking water source followed by restorative land management actions to reduce those threats. The result of this process produced much more than a planning document but engaged multiple community members in a process of empowerment and self-determination. The process of plan-making produced many unintended results including human–land connectivity, reconnection with the water spirit, as well as the reclaiming of indigenous planning. Source water protection planning may not correct all the current water system inadequacies that exist on many First Nations, but it will empower communities to take action to protect their drinking water sources for future generations as a pathway to local water security.

**Keywords:** water security; Canada; Saskatchewan; First Nations; drinking water; source water protection planning; colonization

## 1. Introduction

In Canada, the Indian Act of 1876 [1] enabled the creation of "lands reserved for the Indians" also known as 'Indian Reservations'. Indian Reservations in Canada are noncontiguous and isolated from municipal towns and villages, often located away from rivers and open bodies of water. The Indian Act [1] so served to outlaw cultural traditions, restrict individual mobility, and disrupt indigenous language. Collectively, these and other forms of colonial authority such as government funded and church operated residential schools aimed to erase all things 'Indian' from the Canadian 'settler' landscape, a state-organized process recently described as 'cultural genocide' [2]. The impact of

the Indian Act [1] on the Indigenous people in Canada was, and continues to be, profound and beyond the scope of this paper. Instead, we direct our focus on the degree to which the 'Indian Reservation' (Reservation) system contributes to a general lack of water security for Indigenous people. We define water security as "sustainable access, on a watershed basis, to adequate quantities of water, of acceptable quality, to ensure human, and ecosystem health" [3]. The term 'Indigenous' in Canada includes First Nations, Métis, and Inuit people. In this paper, we describe a water planning process and how it aims to improve water security with a First Nation community in the Canadian province of Saskatchewan. We use the terms "Reservation" and "First Nation community" interchangeably and synonymously in this paper.

The paper begins with an overview of Canada's water governance framework. Next, water security issues as experienced in First Nation communities will be described. The subject of source water protection, and in particular the multiple benefits of community-based water planning, will be introduced as a pathway to greater water security in First Nation communities.

## 1.1. Water Governance

Water governance in Canada is highly fragmented across multiple layers of government departments and agencies [4]. Canada's Constitution Act of 1982 [5] assigns responsibility over water resources to the provinces with each province (10 provinces in total) having separate water governance rules and regulations. In addition, authority over watershed planning in many regions has been further devolved to local watershed organizations that operate in isolation not only from one another but from First Nation communities within the same watershed. First Nation reservations (over 700 in Canada) fall under federal authority in the Constitution Act [5]. As a result, water resource management in First Nation communities is a federal responsibility. These jurisdictional boundaries result in fragmented governance structures, inconsistent planning and management programs, and a patchwork of drinking water quality regulations and standards [4]. The development of water resources and the expansion of resource extraction activities continue today in many parts of Canada without prior and informed consent from First Nation communities [6]. These developments and associated impacts range from damned rivers and flooded valleys to polluted surface water and contaminated groundwater [6,7].

## 1.2. The Water Problem

The impact of state-led water resource development combined with often poor raw water quality and fragmented water governance regimes contribute to poor water quality in many Reservations [7,8]. As of June 2018 there were a combined total of 85 'Drinking Water Advisories' and 'Boil Water Advisories' in effect on Reservations south of the 60th parallel [9]. At any one time, roughly one in seven Reservations in Canada is advised not to drink household tap water. A drinking water advisory is a preventative measure to protect public health from confirmed or suspected microbial or chemical contamination [10]. More problematic is an emergency boil water advisory—issued after confirmation of water supply contamination with fecal pollution indicator organisms [10]. These advisories are issued by the First Nation after water quality results are confirmed by an Environmental Health Officer from Health Canada. Boil water advisories are 2.5 times more frequent for First Nation communities than for non-First Nation communities [11,12]. In addition, approximately 30% of First Nation community water systems are classified as high risk systems and the number of water-borne infections in First Nations communities is an alarming 26 times higher than the Canadian national average [11,13]. Small water systems have a higher likelihood of a boil water advisory, largely the result of limited technical and financial capacity, infrequent expert oversight, and increased likelihood of source water contamination [14]. First Nation boil water advisories in Saskatchewan account for more than 25% of all advisories in First Nations across Canada. This disproportionate burden of water advisories may be explained by the small population size of the 70 total First Nation communities in

Saskatchewan. Saskatchewan has 70 First Nation communities each with a total population as reported in the 2011 Canadian census of less than 500 residents [14].

　　While poor water quality is frequently reported in the media [12,13] much less reported are the causes of these contamination events. To trace the origin of these events requires some reflection on the impacts of colonialism on Indigenous People in Canada. To promote European settlement and associated agricultural development in Canada required the removal of Indigenous People from the land. The federal government achieved this through the Indian Act [1] with the creation of Reservations. Unfortunately, many Reservations were created prior to any infrastructure planning such as the installation of piped water distribution systems, water treatment and community sewer systems. The legacy of such poor planning practice is visible today in many First Nation communities. For example, in Saskatchewan only 74 percent of homes are serviced with piped water from a community water distribution system while 21 percent of homes are serviced by truck delivery [15]. The remaining homes are served by private individual wells. Trucked water delivery consists of community owned and operated trucks that are filled with water from the water treatment plant. The water trucks then travel throughout the community to deliver water into household cisterns or water holding tanks using hoses transported on the truck(s). This method of water delivery is prone to drinking water contamination. Water truck tanks are infrequently cleaned owing to busy delivery schedules. In other instances, the water fill hose from the treatment plant may be compromised by airborne contaminants. In addition, the water truck delivery hoses are regularly in contact with the ground during water deliveries before entering the household water cisterns. In Saskatchewan, approximately one in three truck delivery communities is on a boil water advisory at any one time [15]. Added to this is the high cost of truck water delivery in both operation and maintenance of truck fleets as well as driver salaries.

　　Household sewage disposal is another potential source of drinking water contamination as well as public health concern. A common form of household sewage disposal is a 'jet-out' pipe exiting a home and transferring raw sewage into a backyard area, often directly onto the ground surface. In Saskatchewan, approximately 43 percent of on-reserve households are serviced with a 'jet-out' system of effluent disposal [15]. Approximately 50 percent of the homes are on a piped community sewage system while the remaining 7% are on a truck-haul system [15]. Both the piped community sewage systems and the truck-haul sewage are discharged into constructed sewage lagoons. Most sewage lagoons in First Nation communities are not lined with an impermeable barrier and therefore may contribute to groundwater contamination. The majority of First Nation communities in Saskatchewan (75 percent) source their drinking water from groundwater supplies [15]. Landfills are another threat to groundwater contamination. Most landfills in First Nation communities consist of large open pit excavations accepting all forms of household and commercial solid waste material. These landfill pits remain unregulated by all levels of government authority, including the federal government, allowing potential contaminants to enter the groundwater. In all these examples, the federal government, through its assigned departments, agencies, and ministries, designed and financed these infrastructure projects as a means of establishing permanent settlement on Reservations. We suggest here that the current lack of water security in First Nation communities was created and perpetuated by various state institutions of the federal government.

*1.3. Source Water Protection Planning*

　　Source water is untreated water from groundwater or surface water sources that supplies drinking water for human consumption. Source water protection is a vital first step in the protection of water supplies, often referred to as the first step in the multibarrier approach to safe drinking water [16]. The multibarrier approach (MBA) to clean drinking water is "an integrated system of procedures, processes, and tools that collectively prevent or reduce the contamination of drinking water from source to tap in order to reduce risks to public health" [16]. The goal of the MBA in drinking water management is to reduce the risk of drinking water contamination through the presence of system redundancies, or barriers, built into the water system. The Canadian Council of Ministers of the Environment [17]

described three main components in the MBA beginning with protecting the source of water from the threat of contamination. The second barrier is the treatment of drinking water through various methods including chlorination, filtration, as well as other chemical and mechanical treatments. The third barrier is maintenance, monitoring, and testing of the water distribution system.

Source water protection planning offers a means of addressing past and present land use activities that negatively impact drinking water quality and human health [18,19]. Through source water protection planning there is opportunity to not only foster community health intervention through water security but also to 'reclaim' Indigenous planning [20]. The reclaiming of Indigenous planning alludes to the long tradition of land use planning by Indigenous Peoples prior to European colonization. The purposeful act of planning, or land use planning, is nothing new to Indigenous People. Decisions regarding the timing of hunt and harvest and the location of trap lines and fish nets required intimate knowledge of both land and water [20,21]. Planning was a core component of everyday life for Indigenous People long before colonization. In the words of Matunga [20], "Indigenous planning has always existed. Indigenous communities predate colonialism and were planned according to their own traditions and sets of practices."

Here, attention is drawn to the relational components of kinship, custodial territory, traditional knowledge, cultural beliefs, and intergenerational considerations as foundational building blocks to Indigenous planning [21]. The principles of Indigenous planning center on the linkages between the holistic components of the natural, spirit, and human world [21,22]. Indigenous relation to water (and life) is the understanding that water is not merely associated with life, or a part of life, but that water is life [7]. In the words of Castleden et al., the health of an individual must take into consideration the health of the immediate custodial territory [23]. In this context, healthy water and land is more than a determinant of a healthy individual but of a healthy relationship with Creator.

## 2. Methods

The study site for this source water protection planning process was Muskowekwan First Nation in Saskatchewan, Canada. Muskowekwan First Nation is a Saulteaux (Ojibway) First Nation located in Treaty 4 territory approximately 140 km northeast of Regina, Saskatchewan (see Figure 1). The registered population of Muskowekwan First Nation is approximately 1800 with 500 members living on reserve. Initially, the water treatment plant operator from Muskowekwan First Nation invited the lead author to the community to provide a presentation on the benefits of source water protection to leadership (Chief and Council). Following that presentation leadership decided to engage with the lead author on a community-based source water protection planning process. The planning process and community meetings that ensued followed Ownership, Control, Access, and Possession (OCAP) principles [24], and was approved by the University of Saskatchewan Research Ethics Board.

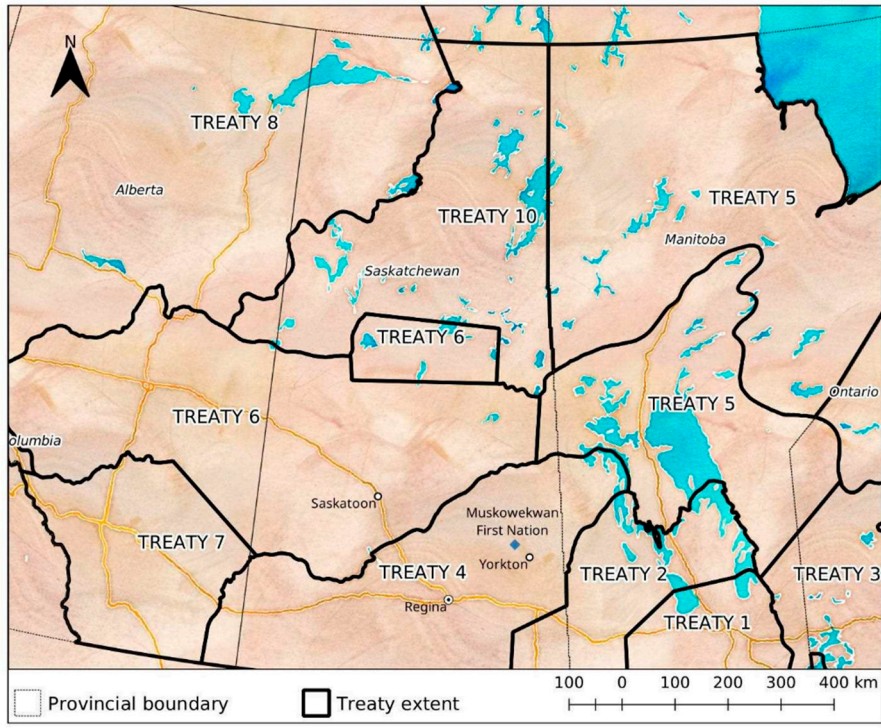

**Figure 1.** Muskowekwan First Nation, Treaty 4, Saskatchewan.

*2.1. Planning Framework*

The planning process used at Muskowekwan First Nation was adapted from an established five-stage source water protection planning framework [25]. The federal government's planning framework represents a structured, rational planning approach with the specific purpose of assisting First Nations in Canada with drinking water protection. Modifications were made to the framework to provide more space for relationship and trust-building between researchers and community. This modified planning framework (see Figure 2) provides greater opportunity for inclusion of First Nation values and perspectives on water as well as space for open dialogue with community members, particularly with Elders. In addition, the adapted framework recognizes community protocols including but not limited to an opening and closing prayer, tobacco gifting to Elders and lunchtime meal. The community meetings were facilitated by the graduate student, Kellie Grant. The following describes the five stages of the planning framework used in this study.

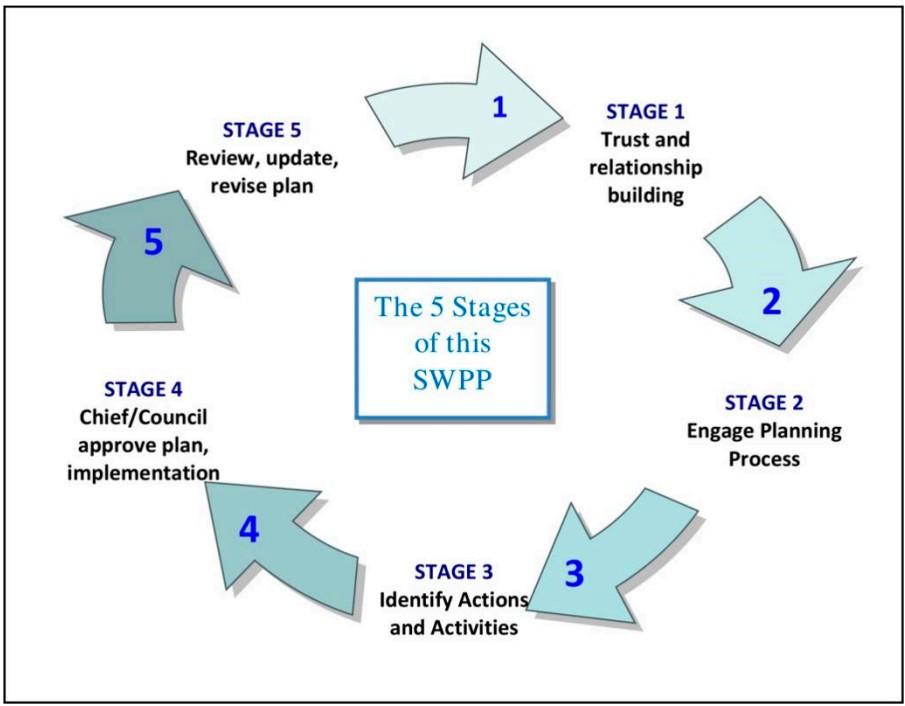

**Figure 2.** Planning framework.

### 2.1.1. Trust and Relationship Building (Stage 1)

The first stage of the plan-making process involves trust and relationship building between researchers and community. This is an opportunity to share backgrounds and experiences related to source water protection planning, management, and research. The water-related research needs of the community and the key research questions should be discussed at this first stage. Past research in First Nation communities was not always respectful of local culture or beneficial to local communities [23]. The working committee size and composition should be determined at this stage to include Elders, youth and representatives from key departments of the First Nation including Health, Lands, Water and Finance. At Muskowekwan, the working committee was chaired by the water treatment plant operator who also became the 'Plan Champion'. Other members included representatives from the Health Center, financial administration, lands administration, an Elder, Youth, an elected official, Muskowekwan school staff, and the water truck operator. It was key to gather a range of people with experience and knowledge of the water system and potential threats to the water sources. All meetings were held in council chambers in an open format allowing other community members to attend and engage at any time. This was advantageous as many ideas and experiences could be openly shared. This approach further built trust and reciprocity between and among those engaged in the plan-making process.

### 2.1.2. Engage Planning Process (Stage 2)

The second stage of the planning framework aims to engage community members and leadership along with the researchers in the plan-making process. An assessment of the water system was undertaken at Muskowekwan to gather information relating to the source of water, location of water intake, type and age of water treatment and distribution system, extent of service area, as well as the number of residential, commercial, and institutional users served by the water system. Following the description of the water system an inventory of potential contaminant sources was undertaken. Using local knowledge, the working committee developed an inventory of all known or perceived land uses and activities with potential to degrade water quality. This inventory included all potential human-generated sources of contamination as well as all natural sources of contamination.

The final component of the source water assessment is a quantitative risk assessment of known and perceived threats to the water source. Risk is defined here as the likelihood of an occurrence multiplied by the potential impact of the occurrence (see Figure 3). Both the likelihood and impact of occurrence range in numeric value from 1 (most unlikely and insignificant, respectively) to 5 (almost certain and catastrophic, respectively). The final risk value assessment for each identified threat will range from 1 (lowest) to 25 (highest).

| Likelihood of Occurrence | Impact of Occurrence | | | | |
|---|---|---|---|---|---|
| | Insignificant 1 | Minor 2 | Moderate 3 | Severe 4 | Catastrophic 5 |
| Most Unlikely 1 | 1 Low Risk | 2 | 3 | 4 | 5 |
| Unlikely 2 | 2 | 4 | 6 | 8 | 10 |
| Likely 3 | 3 | 6 | | 12 | 15 |
| Probable 4 | 4 | 8 | 12 | 16 | 20 |
| Almost Certain 5 | 5 | 10 | 15 | 20 | 25 High Risk |

**Figure 3.** Risk matrix.

### 2.1.3. Identify Community Actions and Activities (Stage 3)

Upon completion of the risk ranking (Stage 2), the working committee focused attention on the many community actions and activities to address the identified risks to source water. Management actions and activities are intended to reduce, or eliminate, identified risks to source water. Local knowledge and experiences of the working committee is critical in this phase. Where possible, the timing of start and projected completion of all actions and activities was noted in the plan along with potential funding sources and necessary partnerships to complete each action and activity.

### 2.1.4. Community Approval of Plan (Stage 4)

In this stage, leadership was asked to comment on the draft plan. After text revisions and the addition of art work, Elder stories, youth input, and photography, the plan was presented for approval to leadership by the working committee and the university researchers. In addition, this is a time when the plan can also be shared with social media, community radio, provincial and local media, all stakeholders, as well as government and industry partners for information. Implementation of actions and activities will follow under direction of the plan champion and community leadership.

Plan implementation commenced immediately at Muskowekwan First Nation. Priority actions from the plan were targeted by the working committee and included a cistern cleaning program (Figure 4), well decommissioning, wellhead protection (Figures 5 and 6), and remediation of a contaminated site. Approximately 20 cisterns were prioritized for manual cleaning in 2015, with another 10 cisterns to be cleaned annually. A large diameter well was decommissioned protecting against groundwater contamination. Wellhead protection was prioritized to protect against floodwater infiltration, surface contamination and grass fire protection.

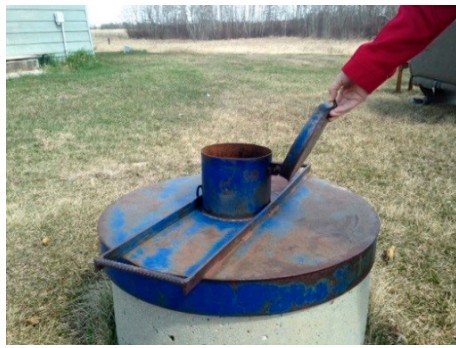

**Figure 4.** Household water cistern.

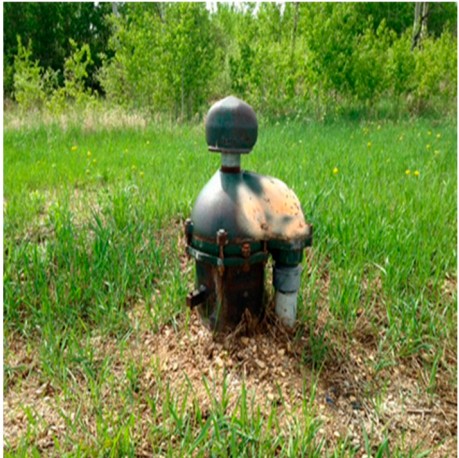

**Figure 5.** Unprotected community well-head (vegetation encroachment, no mounding).

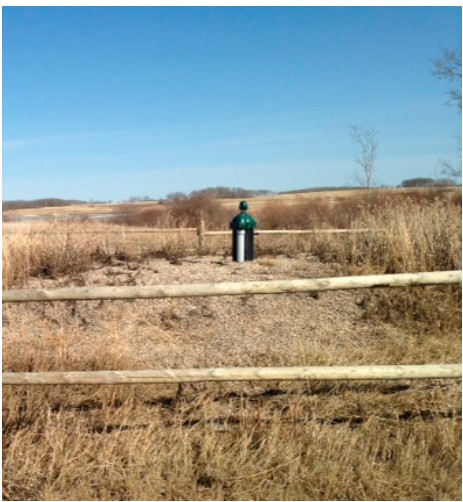

**Figure 6.** Protected community well-head (extended well casing, fence, gravel mounding).

2.1.5. Plan Review and Update (Stage 5)

A full review of the plan by the working committee is recommended on an annual basis with the purpose of reporting plan implementation progress to leadership and the community at large. A benefit of annual plan review includes sustaining momentum to implement the plan, opportunity to celebrate success from the plan, and opportunity to adjust or update the plan based on any new information. Reporting to the community can take many forms including local radio, social media, newspaper, and council reports, but also at community celebration events such as Treaty Days.

## 3. Results

Trust and relationship building between university researchers and the First Nation was an essential first stage for this planning process. This was initiated at the water workshops hosted by Muskowekwan with the working committee and community members. The planning process consisted of six meetings over a twelve-month period involving the university researchers and the working committee.

During the second stage of the planning process details of the water system, as well as risks to the water supply, were discussed and shared between and among all participants. The community water distribution system at Muskowekwan First Nation draws from two groundwater wells each located next to the water treatment plant. The majority of households (approximately 90) receive potable water by truck delivery (trucked water) into household cisterns from the Water Treatment Plant. The school, band office, health center, and 50–60 households are serviced by a piped distribution system directly from the water treatment plant. Five to six households obtain their water from individual wells. Through collaborative discussion, there was agreement regarding identification of risks to the water source. Risk ranking was determined using the risk matric previously described (see Figure 3). All potential contaminant threats, risk ranking, and contaminant of concern are listed in Table 1.

**Table 1.** Potential water contaminants at Muskowekwan First Nation.

| Contaminant Threat | Risk Ranking (score 20–25) | Contaminant of Concern |
|---|---|---|
| Inadequate sewage lagoon cap. | 25 (high) | Sewage effluent, *E. coli* contamination |
| Private wells on reserve | 25 | Poor water quality, bacterial contamination |
| Former railway/industrial lands | 25 | Diesel fuel spills, creosote contaminants |
| Cisterns | 25 | Infiltration, cracks in concrete, broken necks |
| Septic 'shoot-outs' | 25 | Improper disposal, sewage in backyards |
| Abandoned, uncapped wells | 25 | Surface to groundwater contaminants |
| Household heating fuel tanks | 25 | Leaking tanks, oil contamination |
| Unauthorized waste disposal | 25 | Unknown materials, contaminants |
| Train derailment | 25 | Oil, petroleum products, chemicals |
| Overland flooding | 25 | Mobilization of contaminants |
| Abandoned houses | 20 (medium) | Building material breakdown |
| Abandoned vehicles | 16 | Vehicle fluids, batteries |
| Illegal dumping | 15 | Batteries, appliances, fuel tanks, tires |
| Animal carcasses | 15 | Bacteria, animal waste |
| Agriculture –TLE Lands | 15 | Chemical spray, fertilizer, pesticides |
| Horses, Dogs | 12 | Animal waste. rodents |
| Diesel Shed | 8 (low) | Diesel fuel, chemicals |
| Backyard mechanic | 8 | Dumping oil, liquid contaminants |
| Macza Lands (former feedlot) | 5 | Chemicals, oils, storage |
| Landfill sites (unlined) | 5 | Open dumping areas, propane tanks, batteries, tires, animal waste, appliances |
| Transport trucks | 4 | Potential highway fuel spills, hazardous goods |
| Former hide plant | 3 | Animal waste; hides |
| Lambert and sewage lagoon | 3 | Sewage effluent |

The human health impacts of the federal government's water infrastructure approach on 'Indian Reserves', a system rife with technical and design system flaws continues to produce poor drinking water quality. Through a collaborative source water protection planning process two significant outcomes were revealed. The first outcome was risk aversion through adaptation strategies and management actions to protect drinking water sources. The working committee quickly identified what they felt were appropriate adaptation strategies and management actions (see Table 2). Implementation of the management actions commenced immediately after the plan was approved by leadership. The first priorities for plan implementation included cistern cleaning and repair. This was achieved using local knowledge of cistern age and condition. The plan champion and water treatment plant operator prioritized annual cleaning and repair of those cisterns most in need of urgent attention. Federal government funding supported these repairs in year one and two after plan completion to bring all cisterns into a state of satisfactory repair as determined by the plan champion, Mr Julius Manitopyes. Well-head protection for the community's two main supply wells was another priority project. The extension of the well head, gravel berm, and fencing was all coordinated by the plan champion and expensed to the same federal government funding program. Other management actions included decommissioning a large bore diameter well in the community and remediation of the diesel shed contamination site. What became immediately apparent upon completion of the planning stage (Stages 1–3, Figure 2) was the need for a coordinator to lead the plan implementation stage (Stage 4, Figure 2). Implementation progress is contingent on the presence of a plan implementation champion. This research has identified the importance of human capacity, in this case a community champion, to advance plan implementation [23]. While a plan champion emerged from the working committee to guide development of the plan on a volunteer basis, plan implementation requires a longer, multiyear commitment. This could be addressed by leadership through assignment of work duties. The Lands Manager for the First Nation community would be an ideal person to undertake the role of plan implementation coordinator. Table 2 illustrates management actions as well as the responsible agencies and potential funding sources for each of the highest ranking contaminant threat only. Of course, in the final plan this detail of information is provided for all contaminant threat sources—high, medium, and low risk.

The second outcome was the reclaiming of indigenous planning through the practice of source water protection planning. In combination, these two outcomes contribute positively and directly to building water security. The planning process served to strengthen cultural identity as working committee members and the community-at-large openly discussed the cultural and spiritual importance of healthy water. The protection of water and land through plan implementation directly contributes to local capacity for action [25], self-determination, and community empowerment. Source water protection planning and indigenous land use planning, in general, has an important role to play in building indigenous water security [26]. In particular, the extension of indigenous water security to include nonmaterial metrics such as, for example, respect for water, water as medicine and water as a life form, is gaining prominence in the literature [6,27]. This is not to exclude the more conventional metrics of water availability, quantity and quality, but rather, to expand the definition of water security to be more inclusive of other cultural values that attach to water [27].

**Table 2.** Contaminant threat, management actions, and funding sources.

| Contaminant Threat | Risk Ranking (Highest Score *) | Management Actions | Responsible Agency/Funding Source |
|---|---|---|---|
| 1. Inadequate sewage lagoon capacity | High 25 | Build new lagoon, partner with municipality and new industry | Lestock Council; Muskowekwan; Federal Gov't |
| 2. Private wells on reserve | High 25 | Chlorination shock treatment, post 'do not consume' advisory, household treatment | Muskowekwan; Sask Water Security Agency |
| 3. Former railway, Industrial lands | High 25 | Site assessment, contaminant remediation | Muskowekwan; Railway Company |
| 4. Cisterns | High (25) | Initiate annual cistern cleaning and repair, neck extension on each cistern, protect cistern; Convert to water distribution system | Muskowekwan Federal Gov't |
| 5. Septic 'shoot-outs' | High (25) | Extend outflow pipe from house Upgrade to in-ground septic fields | Muskowekwan; Federal Gov't |
| 6. Abandoned, uncapped wells | High (25) | Identify all well locations, cap wells | Muskowekwan; Sask Water Security Agency; Federal Gov't |
| 7. Household heating fuel tanks | High (25) | Collect all tanks for disposal, store at landfill | Muskowekwan; Federal Gov't |
| 8. Unauthorized waste disposal | High (25) | Inform all contractors to remove waste materials. Education, signage | Muskowekwan; Federal Gov't |
| 9. Train derailment | High (25) | Emergency response training; remediation after a spill | Muskowekwan; Railway company |
| 10. Flooding | High (25) | Increasing rain events, climate change. Flood adaptation, sand bagging sensitive areas such as lift station; monitor flood prone areas; flood awareness. | Muskowekwan; Sask Water Security Agency; Federal Gov't |

* Highest risk ranking (25) reflects maximum likelihood (5) multiplied by maximum impact (5).

## 4. Conclusions

The colonial domination over Indigenous Peoples in Canada created a chain of events that continues to impose negative health impacts on First Nation communities. In this paper, examples of poorly designed community infrastructure services have been identified as legacy projects that reproduce undrinkable water in many First Nation communities. These community infrastructure services include inadequacies in housing, water supply and delivery, wastewater disposal, and solid waste management. In the absence of sustainable infrastructure human health will remain at risk measured by persistent boil water advisories and "do not drink" orders issued by health officials.

In response to this legacy of poor infrastructure planning, First Nation communities are now building local capacity to reclaim indigenous planning with the uptake of source water protection plans. Indigenous planning, in particular source water protection planning, has wide potential as a means toward greater water security. The planning process itself served to facilitate dialogue, trust-building, and reciprocity between participants and in particular between community and university researchers. In addition, this planning process helped to build local capacity to undertake water (and land-based) planning and management. We see potential for source water protection planning to make a positive contribution to both the material (water quality and quantity) and nonmaterial (social and cultural) dimensions of water security in First Nations.

**Author Contributions:** Conceptualization, R.J.P. and K.G.; Methodology, R.J.P. and L.B.; Validation, L.B. and K.G.; Investigation, R.J.P. and K.G.; Writing—Original Draft Preparation, R.J.P.; Writing—Review and Editing, K.G. and L.B.; Supervision, R.J.P.; Project Administration, R.J.P.

**Funding:** This research was funded by Canadian Pacific Railway Partnership Program in Aboriginal Development.

**Acknowledgments:** The authors are grateful to the Working Committee for their contributions to the development of the Muskowekwan First Nation Source Water Protection Plan, and in particular, Julius Manitopyes, Plan Champion, and Water Treatment Plant Operator for Muskowekwan First Nation. Map 1 produced by Warrick Baijius, PhD Candidate, University of Saskatchewan.

**Conflicts of Interest:** The authors declare no conflicts of interest. The funders had no role in the design of the study; in the collection, analyses, or interpretation of data; in the writing of the manuscript, or in the decision to publish the results.

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
