# Peer review of "Reclaiming Indigenous Planning as a Pathway to Local Water Security"

_water, doi:10.3390/w11050936_

Round 1

Reviewer 1 Report

It was interesting to read about the treatment of First Nation people in Canada. However, the description of the project undertaken is rather vague, and a bit difficult, for an outsider like this reviewer, to follow. For example, it is stated that the total size of the population living in First Nation reserves is around 500. Then, we learn that the project deals with one that has a population of 500.

What are we supposed to learn from Fig. 1? It does not tell me anything that I can make sense of, except that is shows a scattered First Nation population. There is nothing in the paper itself, that indicates that the target population was scattered in a number of smaller settlements.

The context is poorly described. Who was on the Working Committee? Were there any conflicts between the Canadian water management and that of the project community? What was the Touchwood Agency Tribal Council in 2011? How was the First Nation community organized. There seems to have been a school, a band office (what is that?) and a health centre. In addition, there was obviously a water treatment plant, but how was the water treated?

The work on the ground is poorly described. What were the criteria for deciding whether a cistern should be cleaned or replaced and whether a well should be decommissioned or get wellhead protection. Where did the funds come from? How was the “Indigenously planned” water safety system different from that of a WHO water safety plan?

Author Response

Hello, thank you for your comments, they have been very helpful. Please note our responses are in CAPS:

1. It was interesting to read about the treatment of First Nation people in Canada. However, the description of the project undertaken is rather vague, and a bit difficult, for an outsider like this reviewer, to follow. For example, it is stated that the total size of the population living in First Nation reserves is around 500. Then, we learn that the project deals with one that has a population of 500.

To further explain, the point made is that the vast majority 70 of 76 communities have a population of under 500 people. This places a burden of infrastructure and financing in so many small and isolated first nation communities.

The description of the project have been given greater clarity (see 39-58, various revisions).

2. What are we supposed to learn from Fig. 1? It does not tell me anything that I can make sense of, except that is shows a scattered First Nation population. There is nothing in the paper itself, that indicates that the target population was scattered in a number of smaller settlements.

We fel the legend in figure 1 is self explanatory the yellow star indicted the location of the study. The scattered red is only to show the spatial distribution of first nations in saskatchewan as a general information guide.

3. The work on the ground is poorly described. What were the criteria for deciding whether a cistern should be cleaned or replaced and whether a well should be decommissioned or get wellhead protection. Where did the funds come from? How was the “Indigenously planned” water safety system different from that of a WHO water safety plan?

Thank you. This is a helpful comment.

Table 2 now identifies the responsible agency and potential source of funding for individual management actions and implementation. Cistern repair and well head protection is better described in lines 294-305.the plan champion coordinated these projects on the ground, funding from the federal government.

The source protection planning system used in this study is not dis-similar to that of the world health organization. In fact, it is largely modelled off the usepa model. All these model are similar and follow a rational planning model approach.

Reviewer 2 Report

Thank you for the opportunity to review this paper - I had the opportunity to visit an FN area briefly in Saskatoon during 2018 - so the paper is interesting and relevant. It was a pleasure to read.

I have made a few comments/suggestions in the PDF file for your consideration. Only the one of use of definition of acronyms on first use (OCAP) is, I think, critical; the rest are suggestions to strengthen the paper further.

Author Response

Hello,

Please see comments from reviewer, and response from authors (IN CAPS):

This seems a strange way to refer to a trucking system - perhaps revise to truck-haul unless this is a Saskatoon specific reference?

Error Corrected, Thank You (Lines 120,121)

Please define acronym on first use for readers unfamiliar with the term "ownership, control, access and possession"

Error Corrected, Thank You (Line 174)

Perhaps delete "will" and revise to "...following describes..."

Error Correct, Thank You (Line188)

The highlighted sentence seems to repeat the information in the previous para (sentence beginning Using local knowledge, the working committee aims...). Please consider if this is the case or not - in either case please consider a revision to clarify the paras?

The highlighted senternce has been deleted. Thank you. (lines 222, 223)

Would this be clearer to indicate "Inadequate sewage lagoon capacity"?

The Recommended Wording Has Been Inserted, Thank You (Tables 1, 2)

Did the planning process really only identify high risk category threats? If all risks are categorized as high -then how does the community or planning council select priorities to schedule different interventions other than on cost?  Is it feasible to include a table (or annex) that presents a fuller range of Source Water Risks with the range of Risk assessments generated to put Table 2 into the broader context of risks assessed during the planning process?

This is a very helpful comment. The intent was to only list the highest ranking risks. We have changed this table to include all the risks identified. In addition (see table 2), we have added the risk matrix to provide a viual that describes the risk ranking process with greater clarity for the reader (see figure 3).

The table could also indicate which entity would be responsible for addressing the selected solution(s) individual households, community council, state or federal agency?

A very helpful comment. See table 2. this table only lists the highest risk potential contaminants. As recommended, a fourth column was added to include "responsible agency/funding source". Thank you.  

Finally, if possible, proving an indicative cost for activities/interventions to mitigate each identified risk would add value to the paper.

A very helpful comment. See table 2. this table only lists the highest risk potential contaminants. As recommended, a fourth column was added to include "responsible agency/funding source". Thank you. 

Reviewer 3 Report

Thanks for this paper, which I read with interest. The scope is very specific and it does a good job in unpacking the issue, making an empirical contribution to knowledge. I have some minor suggestions that could further improve the paper:

- Can you please clarify whether the correct term is Reserve or Reservation, and explain the difference?

- Can you please define what the Canadian Praire is?

- In the introduction, I would make comparisons with similar contexts around the world and in history, in particular to the case of Apartheid time in South Africa and to the current Apartheid system in Israel: how does the Israeli system use apartheid laws to strengthen its occupation on the occupied Palestinian Territories in the West Bank and Gaza? How similar is this to the Canadian case (quite similar I would assume)? Read the writing of Mark Zeitoun, Jan Selby, and Clemens Messerschmidt on the topic.

- Somewhere in the introduction, I would suggest to speak about the issue of water scarcity, and how water scarcity is socially constructed and it is not a natural issue. This could be put in the beginning of the section on "water problem". You could say something on the lines of:

"Water scarcity, crisis, and problems are not 'natural' issues, but they are often due to mismanagement. It has been argued that they can be socially constructed through discourses to drive towards and open up certain solutions and policy options. In fact, they can be constructed and deployed to put the blame on certain groups in the society, and to include/exclude certain actors from access to water resources" and reference the following readings:

https://www.sciencedirect.com/science/article/abs/pii/S0305750X01000870

https://journals.sagepub.com/doi/abs/10.1068/a45442

https://www.sciencedirect.com/science/article/pii/S1462901118303137

https://www.tandfonline.com/doi/abs/10.1080/20581831.2017.1379493

https://www.tandfonline.com/doi/abs/10.1080/02508060.2017.1344817

In this way, the paper would be able to provide a better context to the overall picture. Hope this helps. 

Author Response

Can you please clarify whether the correct term is Reserve or Reservation, and explain the difference?

Thank you. We have deleted reference to reserve and now only use 'reservation'. We have also explained in the introduction that the term "reserve" and "first nation community" will be used interchangeably.

- Can you please define what the Canadian Prairie is?

Thank you. We no longer make reference to the prairie, instead we reference the province of saskatchewan and the treaty area.  

- In the introduction, I would make comparisons with similar contexts around the world and in history, in particular to the case of Apartheid time in South Africa and to the current Apartheid system in Israel: how does the Israeli system use apartheid laws to strengthen its occupation on the occupied Palestinian Territories in the West Bank and Gaza? How similar is this to the Canadian case (quite similar I would assume)? Read the writing of Mark Zeitoun, Jan Selby, and Clemens Messerschmidt on the topic.

Thank you. This is not a comparative project. We feel the introduction of international cases and situations would distract from the purpose and intent of this paper.

- Somewhere in the introduction, I would suggest to speak about the issue of water scarcity, and how water scarcity is socially constructed and it is not a natural issue. This could be put in the beginning of the section on "water problem". You could say something on the lines of:

"Water scarcity, crisis, and problems are not 'natural' issues, but they are often due to mismanagement. It has been argued that they can be socially constructed through discourses to drive towards and open up certain solutions and policy options. In fact, they can be constructed and deployed to put the blame on certain groups in the society, and to include/exclude certain actors from access to water resources" and reference the following readings:

Thank you. A very interesting suggestion. Please note this paper does not focus on scarcity. The focus is on water security. It would be interesting to examine whether water security has been used as a discursive tool for some other means. Again, this is not the object of this research.